# Wearable Design Requirements Identification and Evaluation

**DOI:** 10.3390/s20092599

**Published:** 2020-05-02

**Authors:** Leire Francés-Morcillo, Paz Morer-Camo, María Isabel Rodríguez-Ferradas, Aitor Cazón-Martín

**Affiliations:** Department of Mechanical Engineering—Industrial Design Area, Tecnun, University of Navarra, 20018 San Sebastian, Spain; pmorer@tecnun.es (P.M.-C.); mirodriguez@tecnun.es (M.I.R.-F.); acazon@tecnun.es (A.C.-M.)

**Keywords:** wearables design requirements, wearables evaluation, human–wearables interaction

## Abstract

Wearable electronics make it possible to monitor human activity and behavior. Most of these devices have not taken into account human factors and they have instead focused on technological issues. This fact could not only affect human–computer interaction and user experience but also the devices’ use cycle. Firstly, this paper presents a classification of wearable design requirements that have been carried out by combining a quantitative and a qualitative methodology. Secondly, we present some evaluation procedures based on design methodologies and human–computer interaction measurement tools. Thus, this contribution aims to provide a roadmap for wearable designers and researchers in order to help them to find more efficient processes by providing a classification of the design requirements and evaluation tools. These resources represent time and resource-saving contributions. Therefore designers and researchers do not have to review the literature. It will no be necessary to carry out exploratory studies for the purposes of identifying requirements or evaluation tools either.

## 1. Introduction

Wearable electronics have become a recurrent technology for solving concrete problems in different fields such as medicine, leisure, sports, etc. These devices make it possible to monitor human activity and behavior, so they provide essential data for dealing with specific human needs. Although high-tech approaches have been developed in wearable technologies, most of them have a short life cycle when they are launched in the marketplace and user acceptance is still not as widespread as expected. Why do these high-tech devices have such life cycle problems? Although they are technologically advanced devices, human factors have not been considered in many studies. Human factors or ergonomics is the scientific discipline that studies the interaction among humans and other elements of a system based on psychological and physiological principles [1]. The main objective of this paper is to suggest a list of design requirements based on both human and technical factors.

### 1.1. Significance and Novelty

This paper consists of two main parts that respond to the main two challenges: the first part describes the research method and results for the identification of the requirements and the second part presents the research method and the results of the evaluation procedures.

These contributions are significant in terms of gathering and clustering the existing information provided by different authors into a unique classification. Additionally, both the list of design requirements and the evaluation tools have been validated by practitioners with experience in the field of wearables, which gives rigor to our findings.

### 1.2. Challenges

There are two main challenges: to review the existing design requirements and provide a homogeneous classification and to suggest an evaluation approach for such design requirements. In order to deal with the first challenge, a combination of quantitative and qualitative methodologies have been applied. In order to face the second challenge, the field of human–computer interaction has been analyzed and some theoretical principles have been used to define an evaluation approach for wearable design requirements.

### 1.3. Scope

The contributions of this paper are addressed to design practitioners and researchers, as well as to engineers in charge of wearable design projects. The theoretical principles of human factors and ergonomics have been the base of this study and it has been carried out following both quantitative and qualitative methodologies in order to cluster the findings of the review.

#### Key Achievement and Suggestion to Solve the Problem

In this paper, we answer the main research question that has guided this study:

How can we help designers to consider human factors during the design process of wearables?

The answer is complex and requires several steps. In this study, the authors provide the designer with a list of design requirements and evaluation procedures for the design of wearables. This gives the designer guidance in the design process of wearables. It also provides a roadmap where they can focus and select those requirements according to their specific contexts of use.

The first section starts by reviewing and mapping some studies that consider both technical and human factors in the definition of wearable design requirements. Then, a combined quantitative and qualitative analysis is done to identify the design requirements. Finally, the classification and the definitions of terms are explained and the results are presented.

The second section is formulated based on the results of the first. Firstly, a review of ergonomic evaluation tools is done. Then, based on the insights from the review, the measurability of design requirements is questioned. Thirdly, the goals of the evaluation framework are presented. Finally, we suggest an evaluation approach for wearable design requirements.

Once the wearable design requirements and their evaluation is presented, all the design processes are validated. The validation is formulated to include investigations into the interconnection of insights, experts’ reports and triangulation [2] in order to give rigor to the qualitative study. The validation of both the design requirements and the evaluation procedures is presented together by: (1) creating an expert questionnaire and (2) conducting an expert workshop.

## 2. Review of the Literature

In contrast to other technological devices, such as laptops and smartphones, wearables live on people’s bodies. This fact involves a wholesale change in the way wearables should be designed. The creation of wearables requires specific concepts, techniques and ingredients involving textiles, electronics and software that consider the diversity of potential users and their environments [3]. Thus, successful wearable usability is no longer about providing technical success, but rather about creating an optimal user experience [4].

As an example of this, Cho [5] defined wearable design requirements through a human focus, such as Maslow’s hierarchy of needs. Maslow’s theory provides a clear and strong basis to identify and prioritize services and design worldwide from a human point of view [3]. From this perspective, six different human aspects that should be considered in the design of wearable systems are defined: usability, functionality, durability, safety, comfort and fashion.

Motti and Caine [6] defined principles related to human factors that should be incorporated during the design phase of wearables and they included the term “wearability”, understood as the interaction between the human body and the wearable object. Dunn [7] shared a similar view and understood wearables as “the relationship between a worn technology and the ability or desire of the user to wear it, is a key element in the successful design of wearable technologies”. In the same way, other authors have also focused on the idea of “wearability”, such as Pantelopoulos and Bourbakis [8], who proposed a new perspective for “wearability” in the specific development of sensor-based systems for health monitoring and prognosis. In contrast to the contribution of Motti and Caine [6], and although sharing a similar point of view, the development of this study was merely technological. The same conclusions can be drawn from the study of Mayol et al. [9], which was based on the development of active vision platforms. After their study, other researchers began to use the term wearability. Venere et al. have also analyzed wearability in several works [10,11] and have undertaken research regarding a user-centered approach for designing wearables [12].

Knight [13] also proposed a methodology for assessing the effects of wearing a computer in terms of physiological energy expenditure, biomechanical effects, discomfort due to musculoskeletal loading and perceptions of wellbeing through a comfort assessment. Canina and Ferraro [14] have addressed the importance of considering not only engineering aspects but also the psycho-physical wellbeing of users. Contreras-Vidal et al. [15] have expressed the need to take a human-centered approach to improve interaction and to make wearables more effective, reliable, safe and engaging.

Accordingly, another similar focus aligned with the user- and body-centered research of wearables can be found in Martijn ten Bhömer’s doctoral thesis, “Designing Embodied Smart Textile Services” [16]. The research explores how close-to-the-body products and services can become meaningful to people. Along the line of embodied design, Tomico [17] provided a discussion about the opportunities and challenges of designing soft wearables, applying the notion of personal meaning to different design cases. Furthermore, in terms of design methods, Wilde et al. [18] presented a framework that enables designers to understand embodied design ideation practices.

Another different focus within the framework of user-centered design is the application of Universal Design Principles to wearables, as developed by Tomberg [19]. This focus helps to move the approach to the design of wearables closer to human factors.

In terms of the transparency of data, Andreoni et al. [20] highlighted the non-intrusive monitoring paradigm, which must be employed in order not to affect user behavior and to his/her own daily activities. As can be seen, the study of wearables through a user-centered approach encompasses many different alternatives. The understanding of such principles enriches and humanizes any technologically driven field, such as that of wearables.

However, detailed information as to how requirements get identified and, particularly, how they are evaluated and integrated in the design process has not been undertaken before. The research outlined in this paper aims to provides a classification of design requirements and also to suggest an evaluation procedure for wearables.

Table 1 summarizes the main key messages of the authors across the different studies mentioned above.

Once the literature review was completed, we identified a great research opportunity in the identification of wearable design requirements based on the consideration of both technical and human aspects. The following sections describe the methodology and the main results of the study.

## 3. Research Method for Wearable Design Requirements Identification

Our research method for wearable design requirements identification was been based on the analysis of seven studies that identified design requirements, taking human factors and human–computer interaction issues into account. Although there are only a few studies, they suggest a representative amount of the design principles that are reviewed and analyzed in this section. Firstly, a mind map is drawn, and all design requirements are represented within the ergonomic physical, cognitive and emotional categories. This categorization is commonly used in studies on human–computer interaction [1] and it is useful to identify the lack of design requirements in each ergonomic category. Figure 1 presents the described methodology.

### 3.1. Reviewing and Mapping Wearable Design Requirements

First, the studies containing wearable design requirement identification were reviewed and a mind map was drawn. This mind map shows a list of proposed design requirements divided into the different studies’ foci and then, into the three ergonomic disciplines: physical, cognitive and emotional. This list of proposed design requirements was gathered from seven studies that analyzed the role of human factors in wearables. While researchers have proposed several approaches related to technical aspects, human factors are still overlooked [1]. Moreover, the existing studies focus on specific applications. This involves different contexts of use, which is why this study has excluded them from this theoretical framework.

Figure 2 shows the requirements that appear in more than one study. The mind map is categorized by the three types of ergonomics, represented in three different colors. The contributions by the different authors are shown inside the categories. The dashed lines represent connections between different groups that have similar meanings.

As can be seen, “comfort” (four out of seven studies) is the most popular category, followed by “intuitiveness” (three out of seven studies) and “movement” (three out of seven). The rest of the design requirements are used twice in the selected studies.

As Figure 2 shows, all the requirements are classified in terms of physical, cognitive or social ergonomics. Although all the studies considered human factors, only a few of them considered requirements that fall within all three domains. Most of the design requirements found in these studies were related to physical ergonomics, and only a few of them were about cognitive and social ergonomics. This means that there is a lack of the system and that the entire design cycle.

As far as design requirements are concerned, the study resulted in a list of 52 different requirements. Looking at the number of requirements within each group, it can be seen that there are many more requirements included in physical ergonomics (30/52) than in cognitive ergonomics (15/52) or emotional ergonomics (7/52).

This fact has nothing to do with their importance, but it may be related to the ease of measuring such requirements. Moreover, each group is not independent, but rather they are intrinsically related to each other—That is, non-compliance of any requirement in a group would directly affect the other groups (e.g., the lack of comfort in one product could affect a user’s decision making or satisfaction).

Moreover, the different terms used interfere with the analysis of each requirement’s weight. In the case of the term “comfort”, some studies consider only physical ergonomics, but others include aspects such as “emotion” [12], so the use of the word is deeper in some cases than in others. This leads to the conclusion that some works use the same term to refer to different concepts. This make it impossible to draw conclusions about the most popular requirements without a specific methodology.

### 3.2. Quantitative Analysis

In order to analyze design requirements, it is important to consider not only the design requirements but also their definition. As previously mentioned, a single term for a design requirement can have different definitions in other studies. This means that, in order to undertake an in-depth analysis, the definition should also be considered. To that end, this study has established the following criteria:-The number of times a specific design requirement is used in the studies (N);-The number of requirements inside the definition (P);-The number of times the requirement appears in another definition or requirement (C).

Table 2 shows the definitions considered in this study for the terms used above.

This section is explained in more detail in Section 9 (Supporting Information).

### 3.3. Qualitative Analysis

The aim of the qualitative methodology is to establish relationships between different design requirements. Doing this it will make it easier to create groups and to analyze cause–effect relationships between different terms and to identify which terms have greater interconnections.

Two techniques have been used in this step: symmetric and asymmetric clustering matrices, which are specific design tools for dealing with large amounts of data and showing interconnections [21].

The development of symmetric and asymmetric clustering matrices is explained in detail in Section 9 (Supporting Information).

## 4. Results from Wearable Design Requirements Definition

Based on the methodology followed for the identification of design requirements, we can conclude that is it not enough to quantify the relevance of terms in different studies, but that it is necessary to analyze the terms’ meanings. Additionally, as can be seen, the design requirements proposed by Cho all have a strong or medium correlation with the rest of the terms. Taking this into account, the classification model will be based on the reference model suggested by Cho. Thus, it includes the design requirements that produced high scores in the symmetric and asymmetric matrices. Moreover, new definitions for such design requirements will be done. In summary, the final design requirements selection will address the following criteria:The classification model will have a homogeneous structure with a balanced distribution within physical, cognitive and emotional ergonomics;The classification model will be based on the reference model proposed by Cho and will be completed by those design requirements that produced a higher score in the symmetric and asymmetric clustering matrices;Due to the strong connection with different terms, the design requirements will be defined as main “design requirements” and others as “sub-design requirements” or “design parameters”;The design requirements and design parameters with similar meanings will be unified.

Based on the criteria described above, Figure 3 presents the suggested classification proposal.

The definitions of the wearable design requirements represented in Figure 3 (wearable requirements wheel model) are given in Appendix A
Table A1.

## 5. Research Method for Wearable Design Requirements Evaluation

In a similar manner to wearable requirements identification, there are few studies that address the issue of wearable evaluation. Because of this, the present section focuses on some studies that have suggested similar evaluation standards in the field of human–computer interaction evaluation [22] that can be reference models for wearables. First, a review of human–computer interaction assessment tools is presented. Then, the relationship between the identified tools and the identified design requirements is analyzed using a measurability analysis. Thirdly, based on the insights of previous steps, the goals of the evaluation framework are defined. Figure 4 presents the described methodology.

### 5.1. Reviewing Wearable Design Requirements Evaluation Tools

Based on the design requirements proposal, this section reviews some of the existing human–computer interaction measurement tools, biomechanical devices and physical work assessment tools. The aim is to link the identified design requirements with the tools and thus to suggest an evaluation approach for testing wearables. In order to establish the relationship between the design requirements and tools, we followed the next procedure:Clustering the evaluation tools into human–computer interaction measurement tools, biomechanical devices and physical work assessment tools;Identifying the main outcomes of the evaluation tools;Comparing the outcomes with the design requirements and analyzing whether or not they fit.

After repeating this procedure with all the outcomes and design requirements, the reviewed tools (see Appendix A
Table A3) can only be linked to some of the design requirements, such as satisfaction, functionality and comfort. The results show that, through the analysis of the reviewed tools, we cannot be sure of the design requirements evaluation. This could be due to the measurability of the design requirements and their degree of development during the design process. Due to the lack of tools associated to the design requirements, the following section will analyze the measurability of such design requirements. In this way, we will be able to determine if wearable design requirements are quantitative or qualitative and thus an evaluation approach based on other kinds of tools and processes can be suggested.

The table that summarizes the biomechanical devices and physical work assessment tools is included in Appendix A as Table A2.

### 5.2. Questioning Measurability of Design Requirements

Analyzing whether methods are quantitative or qualitative is of great interest in the design of wearables. On the one hand, some of the wearable design requirements are quantifiable, yet others are not. That means that some of the design requirements require qualitative approaches to be evaluated. The recent debate over the use of quantitative or qualitative approaches interacts with the debate over objective and subjective measurements. Ergonomics and human factors deal with parameters that are difficult to measure and therefore have no agreement when it comes to the method of evaluation (Table 3). This section analyze whether the identified design requirements are quantifiable or not based on the quantitative and qualitative dimensions defined by Sharples in [1].

Following this procedure, and in order to be able to identify evaluation tools for design requirements, the measurability of such design requirements is analyzed and represented in Table 4.

Although the parameters are quantifiable, they also require an additional qualitative test to test user acceptance. What is indeed a crucial issue in wearables evaluation is to test if wearable design requirements can be verified and validated rather than quantified. Due to these reasons, the measurability analysis is not enough to determine and identify evaluation tools, and the following section presents the goals of the evaluation framework in dealing with this situation.

### 5.3. Goals of the Evaluation Framework

As has been pointed out before, most of the wearable design requirements represent a combination of quantitative and qualitative parameters, so the attempt to measure them through a unique approach does not yield the most accurate results.

So far, some studies have suggested similar evaluation standards in the field of human–computer interaction (HCI) evaluation [22]. As the existing approaches of wearables evaluation are only technical, this section will adapt existing assertions from to human–computer interaction to the field of wearables.

The assertions contemplate the key aspects of evaluation procedures, both for processes and products, and introduce the issues of measurement and a reference model for conducting the assessment. If we take this into account, the suggested assertions for wearables have been modified according to the main differences between human–computer interaction and wearables and, hence, the main differences between their evaluations. Table 5 shows the deducted assertions for wearables based on assertions given in the HCI evaluation.

## 6. Results from Wearable Design Requirements Evaluation Methods

Based on the insights from previous sections, the following conclusions can be obtained:-A unique evaluation tool for each wearable design requirement does not exist;-Design requirements validation is iterative and should be tested through a triangulation of methods;-Prototypes play an important role in such an iterative process.

For this reason, this section presents the evaluation proposal of design requirements based on the integration of different evaluation tools for each design requirement. In order to suggest the evaluation proposal, different evaluations were analyzed and are shown in Appendix A
Table A4 and Table A5.

As can be seen in Table 6, some methods are used several times for different requirements. In this way, and as has been pointed out before, some requirements can be tested at the same time. For example, we could conduct a user trial to test almost all the requirements. Depending on the design stage and level of development, this user trial might give designers different insights.

*In view of this situation*, How Can We Connect Design Requirements and Evaluation Techniques? In order to answer to this question, this section seeks to highlight the role of prototypes. There are different degrees of fidelity in the development of prototypes that can be applied to wearables design. The role of the prototype is different in each design phase. In the early stages, prototypes are mainly used as questions or preference indicators, while in the final stages, they are used as interactivity and functionality indicators [16,24]. In essence, they are one of the key tools for the verification and validation stages. As a result of this view, the first stage insights are related to exploratory and diagnostic results, while the last stage insights show performance measurements [25,26] since prototypes are used to conduct user evaluations and to anticipate the impact of changes [24]. On balance, if the roles of prototypes are important in every design process, they are even more important in such a complex and multidisciplinary field as wearables.

As with prototypes, other design techniques can also be used to evaluate more than one requirement. For this reason, design requirements evaluation tools are not classified according to specific requirements, but to the design process.

As a result of the reality described above, this section has developed new applications for existing design techniques in the field of wearables. It has proposed a comprehensive framework for testing wearable design requirements through a different approach. In particular, a list of techniques that are already used in service or product design is suggested and combined with other existing quantitative methods.

## 7. Validation

This section presents the validation process for both the wearable design requirements validation as well as the wearable design requirements evaluation. Firstly, the contribution of the design requirements wheel model was presented at the International Design Conference (Dubrovnik), where a detailed explanation of the identification of wearable design requirements was presented. International design experts that participated in the conference gave feedback about the methodology and the final classification. Different conclusions about the evaluation procedure were also shared. As a result of this feedback, some aspects of the classification model were modified and some ideas about how to conduct the experts’ interviews were identified. As one example of the classification modifications, some of the design requirements such as “functionality” and “satisfaction” were moved from Level 2 to Level 3. In other words, such design requirements were categorized as general design requirements that contained more specific design requirements.

### 7.1. Survey

#### 7.1.1. Survey Development/Methodology

The survey was developed based on the feedback gathered at the conference. That means that the content of the survey collected the design requirements and different experts were asked to evaluate the suggested classification, as well as to test the list of design requirements based on their experience.

#### 7.1.2. Survey Structure

The survey consisted of four different parts. Firstly, experts were asked about their personal experience of wearables and some of their stories were collected in written diaries. Secondly, a design requirements classification was shown to experts and they had to eliminate or create additional categories. Then, the same classification was extended with subcategories and design parameters and they were asked to create or eliminate categories. Finally, they were asked to suggest specific design or engineering tools which could be helpful to evaluate or test such requirements. The formulated questions in the survey are shown in Table 7.

#### 7.1.3. Participants

Seven different participants took part in the survey. Four of the experts were interviewed on site and three of them were interviewed by electronic surveys.

In the case of on-site surveys, an email invitation was sent. It described the purpose and the main goals of the survey, then the interview was completed face to face.

Regarding electronic surveys, an email invitation was sent and then a telephonic explication was given to assure the recipients’ comprehension of the survey.

Participants were experts on different wearable design phases, from conceptual design and technical design to service design. Thus, different perspectives were gathered. Three of them ran their own business as designers, other one worked in a design consultancy, another ran their own user experience business and the last one worked at a university. Their answers were analyzed one by one and insights were extracted. The repeatability of answers was also considered.

### 7.2. Survey Results

#### 7.2.1. Wearable Design Requirements

As a result, of the first question about wearable design requirements, two experts pointed out that functionality was a synonym of usability, so they would choose only one of these concepts as a design requirement. Other design requirements that were only considered for elimination by one of the experts were durability (due to business strategy), privacy (due to the nature of wearables) and, finally, aesthetics (because they did not consider it a main design requirement). With regard to requirements that must be included, experts suggested interaction, industrialization, price, cognitive overloading, and business design requirements.

Similar results were found for the next question in the survey. One of the experts suggested that we remove life cycle, customization, fashionable, temperature, accuracy, privacy and subtlety and one of the experts suggested that we should add cognitive overloading. Table 8 shows the responses of the seven designers.

#### 7.2.2. Wearable Design Requirements Evaluation

According to the evaluation methods, multiple ideas were gathered. By analyzing the multiple responses, it can be concluded that some of them are the result of the expert consulting a technical datasheet about technical and materials specifications, and others are the result of the expert having carried out a particular experiment, such as an anthropometric or biomechanical study. In this kind of studies, it is necessary to have the user physically present while in experiments, such as testing the performance of a sensor or battery, the physical presence of the user is not required. fact, several user tests in different phases are needed and, in these tests, different prototypes are required. These conclusions are directly linked to some concepts described in Section 1.3, Design Requirements Evaluation. Additionally, the experts suggested some classifications that are useful for this study. The measuring methods suggested by the experts were: checking specifications in technical datasheets, additional experiments and tests and checking existing wearables guidance.

These examples are, in essence, different quantitative and qualitative tools and methods that experts suggest for use during the whole design process. Thus, this insight from the interviews reinforces the reasoning behind the incorporation of a triangulation of methods.

### 7.3. Expert Workshop

A total of twelve participants were involved in the expert workshop: four design researchers (a facilitator and three observers), two ergonomists, two designers, two engineers and two users. For this workshop, two groups were created. Each of the four main expert profiles were distributed equally among the groups. Thus, both groups were composed of an ergonomist, an engineer, a designer and a user and they had to complete the same tasks simultaneously.

The participants were given a design brief to design a smart glove. The tasks they had to complete were categorized as follows: design brief, brainstorming, concept definition, service blueprint, anthropometric study, co-evaluation checkpoint, hardware selection, software selection, hardware architecture, software architecture, prototyping, usability test, concept redefinition.

Once the tasks were finished, participants were asked to complete a co-evaluation checkpoint that contained a list of design requirements. The participants had to specify the current degree of fulfilment of a specific design requirement in the checkpoint. Firstly, they had to quantify the degree of consideration of a specific design requirement from one to five on a Likert Scale. Secondly, they had to analyze if this design requirement was applicable or suitable and explain their reasoning.

The main objective of the expert workshop was to identity iterative points during the design process. Once the workshop was finished, an observer analyzed the co-evaluation checkpoint templates and worked out the degree of fulfilment of each design requirement. Additionally, they also analyzed the rest of the templates to see how far they were modified after completing the co-evaluation checkpoint.

### 7.4. Expert Workshop Results

As mentioned above, the main objective of the expert workshop was to test if having a list of design requirements helped in the design process and, additionally, to identify iterative points. Having taken these objectives into account, the results of the workshop are shown by:-The degree of fulfillment of the design requirements in the first round: The degree of fulfillment is the average of the participants’ responses collected via the checkpoint templates. The aim is to see how having a list of design requirements influenced the design process and thus to see how the participants iterated after completing the checkpoint template;-The degree of fulfillment of the design requirements in the second round: As before, the degree of fulfillment is the average of the participants’ responses collected via the checkpoint templates in the second round. The aim of collecting these values in the second round is to see how the degree of fulfillment increases through the design process;-Design stage influences by the iteration. The design stage, influenced by the iteration, collates the different templates that have been modified after completing the checkpoint template (the list of design requirements). The aim of presenting this information is to see what kinds of modifications the experts made after completing the list of design requirements.

As a result of the process described above, it can be concluded that there is a significant difference when providing a list of design requirements during the design process. The co-evaluation checkpoint was completed by the experts twice: the first was completed after the anthropometric study and the second was completed after the prototyping. As Table 9 shows, there is a considerable difference in the degree of fulfillment of the design requirements between the first round and the second round.

Additionally, the observers analyzed the influence of the co-evaluation checkpoint on other templates. Sometimes, the experts referred to a template that they had already completed in order to change some characteristics. The different templates that were modified are presented in Table 9.

## 8. Discussion and Conclusions

Whenever the issue of wearables design is addressed, widespread doubts about the requirements that they should meet arise. In complex products and hybrid designs, the identification of design requirements is important because of the different factors that influence the design of a product.

In this way, this paper had two main challenges. The first one was to identify the wearable design requirements and the second one was to suggest an evaluation approach for such requirements.

The first challenge was addressed in Section 2 and Section 3. As a result, it can be concluded that the classification of design requirements helps us to cluster design requirements and guide design practitioners and researchers in the design of wearable devices. In terms of the methodology used to identify requirements, a combination of quantitative and qualitative methods was used. Firstly, when the list of requirements was large, a quantitative methodology was applied. In this way, the relevance of terms was considered, then a qualitative methodology was applied. In this phase, the definitions were analyzed by different experts. After presenting our results in the International Conference of Design [25] a report was created and some changes were made to the classifications. Finally, different experts participated in the validation of the classification model. As the results show, the experts that participated in the validation found that the contribution saved them time and was helpful for the design of wearables, taking both technical and human factors into account.

The second challenge was addressed in Section 4 and Section 5. Overall, it can be said that this section opens up a greater discussion as to whether design requirements are evaluable or not. This paper argues that there is not a one-way relationship between a specific design requirement and a specific evaluation technique, but that there are multiple relationships. Some design requirements need specific verification, while others should be validated according to several developmental degrees during the design process. Thus, it can be said that the relationship between requirements and tools is not singular, and that it depends on the context and aims of the project. Depending on the project, some requirements are more essential than others. Additionally, a new debate about the relative applications of quantitative and qualitative approaches has been opened. This has been an issue from the earliest days of design-driven disciplines.

As a result of the reality described above, this paper has developed new applications for existing design techniques in the field of wearables via two key achievements. The first is the classification proposal based on both human and technical factors and the second is the suggestion of a new approach to evaluate such design requirements. Both contributions were analyzed to identify the key stages of the design process where they should be evaluated.

## 9. Supporting Information

This section describes some of the methodological details that were developed in order to achieve the results of this paper.

### 9.1. Methodological Development of Reviewing and Mapping Wearable Design Requirements

#### 9.1.1. Quantitative Analysis

As mentioned in the previous section, the measurement of terms is not enough to give reliable results due to the lack of correlation in the analysis semantics, the generality of the terms and the connection between different terms. For example, the term “comfort” is used in four different studies, and in each one it has a different definition. In some studies, it is used as a design requirement [5,6,11] and in others it is used as a group of requirements [13].

Furthermore, the term “comfort” is included in the definition of other design requirements, which means that a connection could be established. In analyzing the rest of the design requirements, such as “obtrusiveness” or “sizing”, it could also be concluded that they are included within the “comfort” group due to their meaning. These conclusions led us to apply the criteria of quantifying the number of requirements inside the definition (P) and quantifying the number of times a requirement appears in another definition or requirement (C). If we go back to the term “comfort” as an example, we will see that, inside its definition, another design requirement mentioned in another study appears (P = 1). Comfort concerns the freedom from discomfort and pain. Users feeling enough comfort no longer sense the device after wearing it for some time. Comfort involves an acceptable temperature, texture, shape, weight, and tightness. Comfortable devices fit users, enabling normal movements, without constraints.

Flexible materials, for instance, permit normal joint movements. Smaller form factors and more convenient sensor locations on the body can ensure comfort. As this definition shows, the term “weight” is considered a design requirement in other studies [20]. This means that weight is considered a critical factor when measuring comfort. Regarding the number of times “comfort” appears in another definition, it can be seen that there are eight definitions including comfort: “wearability”, “contextual-awareness”, “attachment”, “thermal comfort” (twice), “physiological consideration”. These results show that “comfort” has a great relationship with other terms and definitions.

Figure 3, Figure 4 and Figure 5 show the values of the number of times a specific design requirement is used in the studies (N), the number of requirements inside the definition (P) and the number of times the requirement appears in another definition or requirement (C), which were calculated by analyzing the design requirements and the definitions of the seven reference studies shown in Figure 2. Each figure represents the values for a specific ergonomic domain: Figure 3 shows the values for physical design requirements, Figure 4 shows the values for cognitive design requirements and Figure 5 shows the values for emotional design requirements. Figure 6, Figure 7 and Figure 8 show the values of N, P and C in physical, cognitive an emotional ergonomics consecutively.

This quantitative method helps to set the roadmap for establishing design requirements for wearables, but a qualitative phase is required to complete this first step. Indeed, the quantitative analysis informs us about the connectivity of different terms, but it does not contemplate the connection that two different requirements could have due to their definitions. Thus, the qualitative analysis criteria are detailed below.

#### 9.1.2. Qualitative Analysis

##### Asymmetric Clustering Matrix

The asymmetric clustering matrix is a design tool that compares two entities gathered during research and shows how each set breaks down into clusters based on its relation to the other set. It provides a systematic analysis, facilitates comparisons, reveals patterns and relationships and handles large sets of data, making it easy to visualize. The inputs are two sets of entities based on research findings and a matrix tool for scoring and sorting. The outputs are entity clusters based on the strength of the relationships between them and insights about relationships between two sets of entities.

In our case, one entity is the six design requirements taken from [5], and the other is the 26 design requirements in which N or P or C >1. Thus, the size of the matrix is six by 26, and the scoring criteria are shown in Table 10.

The result of the asymmetric clustering matrix is shown in Table 2, which illustrates the 27 resulting relations. Among the design requirements proposed by Cho [5], the matrix shows that the item “comfort” has 10 relations with the rest of the design requirements proposed in the literature; five of them have a strong relation and five of them have a medium relation. The item “safety” has five relations, four of which are strong and two of which are medium. “Durability” has no relations with the rest of items and “functionality” has only two medium relations. “Usability” has seven relations, of which four are strong and three are medium. Finally, “fashion” has three relations, two of which are strong and one of which is medium.

These results show that the design requirements suggested by [5] do not include most of the design requirements proposed by other authors, and that aspects such as “durability” or “functionality” have no relations or only very few relations.

Within the group of physical ergonomics, only “comfort”, “safety” and “fashion” have a connection with the rest of the items. Regarding cognitive ergonomics, “comfort”, “safety” and “usability” are the only ones that are connected. Finally, within the group of emotional ergonomics, “comfort”, “functionality” and “fashion” are related, but have few connections.

Consequently, although terms from different ergonomic categories are related, a common behavioral criterion cannot be established. The results from the asymmetric clustering matrix are shown in Figure 9.

##### Symmetric Clustering Matrix

Based on the results of the asymmetric clustering matrix, a second matrix was proposed. The aim of the symmetric clustering matrix was to consider more items in order to achieve a higher amount of relations between the design requirements. The design tool is similar to the asymmetric clustering matrix, but it varies in size. In this case, design requirements with N or P or C > 1 were compared with each other, so the matrix had 26 squares. As it is a symmetric matrix, only one half is represented.

Figure 10 shows how the design requirements are related to each other. This resulted in 37 relations, which makes sense when taking into account the fact that more items were used. Although the number of relations is much bigger than in the asymmetric clustering matrix, it can be seen that terms in the same ergonomic category are connected to a greater extent than terms in different ergonomic domains.

In the case of physical ergonomics, “comfort” has the highest number of relations, followed by “form/shape”, “safety”, “wearability” and “harm”. In terms of cognitive ergonomics, “intuitiveness” has the highest number of relations, followed by “usability” and “affordance”. Finally, in the case of emotional ergonomics, only a few relations can be found: “aesthetics”, “customization” and “satisfaction”.

We found nine different design requirements which include other requirements within their definition; “comfort” and “ease of use” are the most commonly used ones, followed by “simplicity” and “satisfaction”. Secondly, there are 12 different requirements that appear in other definitions. Finally, the most common ones are “comfort” followed by “intuitiveness”, “usability” and “movement”. This leads to the conclusion that there are high degrees of interrelatedness and group ability in the design requirements and their definitions.

As can be seen, the results of this matrix are still not firm enough to draw a behavior pattern. Thus, based on the quantitative and qualitative analyses, the main conclusion that can be drawn is that there are no common criteria for selecting and classifying the design requirements.

## Figures and Tables

**Figure 1 sensors-20-02599-f001:**
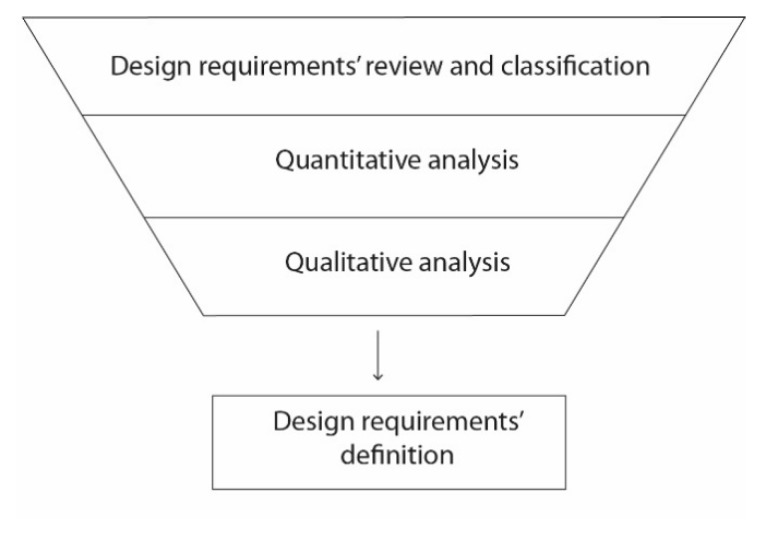
Methodology for wearable design requirements identification.

**Figure 2 sensors-20-02599-f002:**
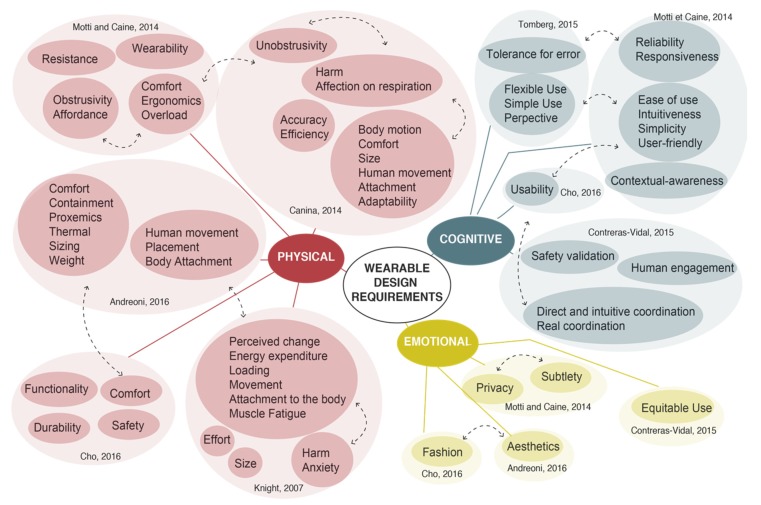
Studies that have considered human factors in the definition of wearable design principles.

**Figure 3 sensors-20-02599-f003:**
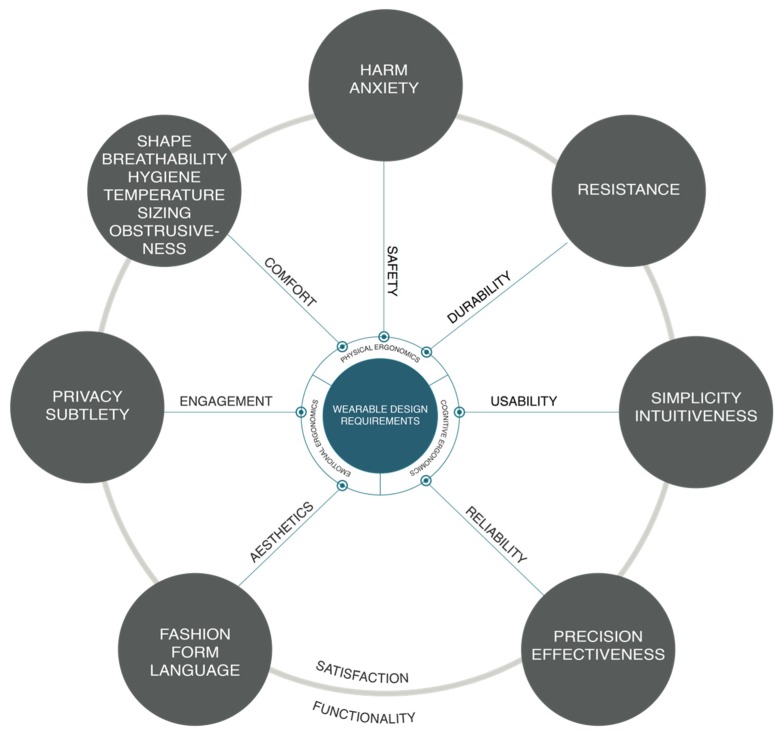
Wearable design requirements wheel model.

**Figure 4 sensors-20-02599-f004:**
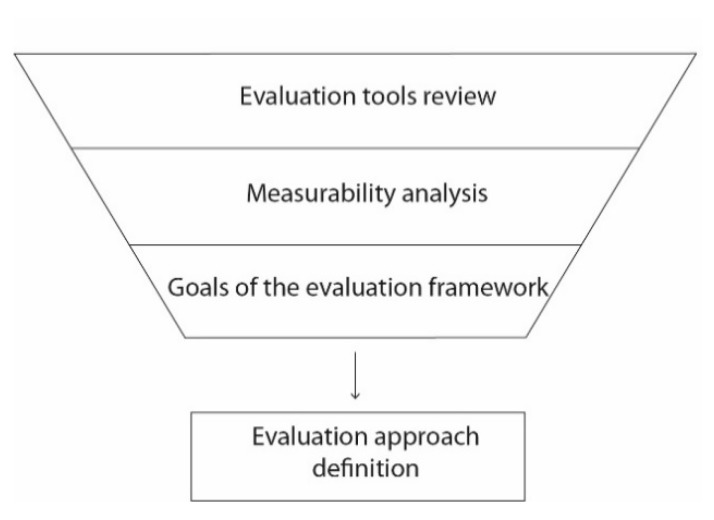
Methodology for wearable evaluation tool identification.

**Figure 5 sensors-20-02599-f005:**
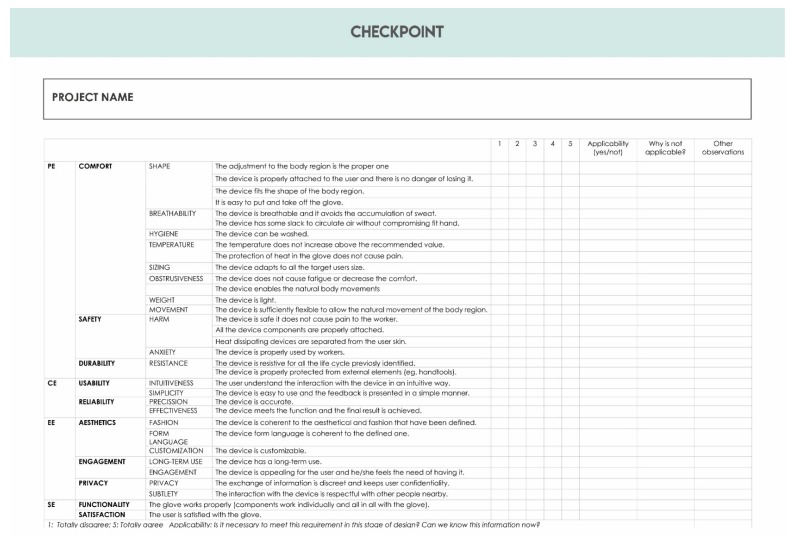
List of design requirements of the co-evaluation checkpoint.

**Figure 6 sensors-20-02599-f006:**
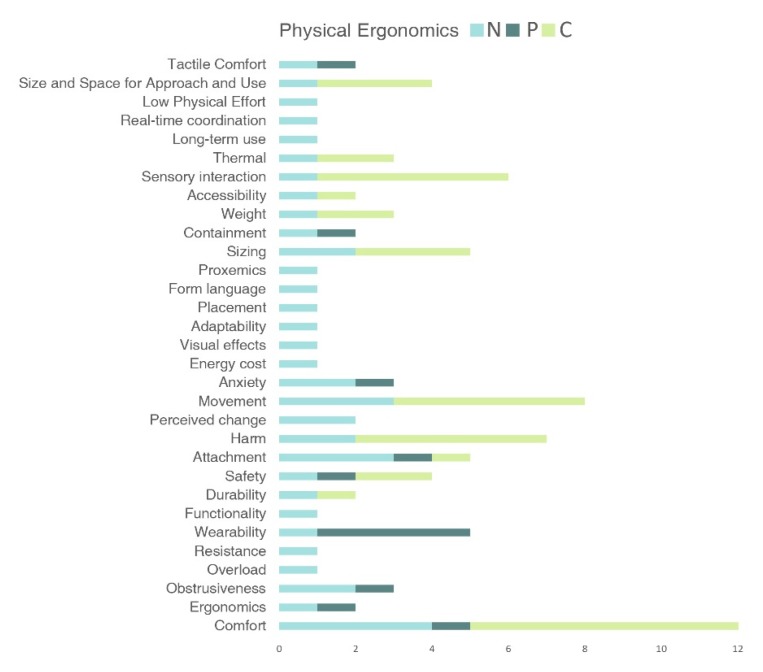
Number of times a specific design requirement is used in the studies (N), number of requirements inside the definition (P) and number of times the requirement appears in another definition or requirement (C) values for physical design requirements.

**Figure 7 sensors-20-02599-f007:**
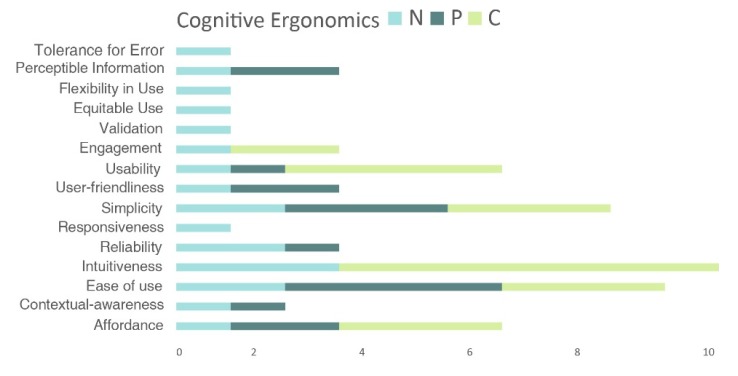
N, P and C values for cognitive design requirements.

**Figure 8 sensors-20-02599-f008:**
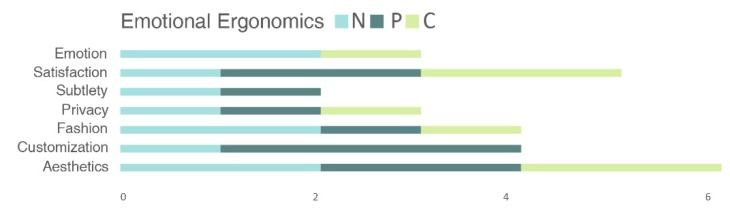
N, P and C values for emotional design requirements.

**Figure 9 sensors-20-02599-f009:**
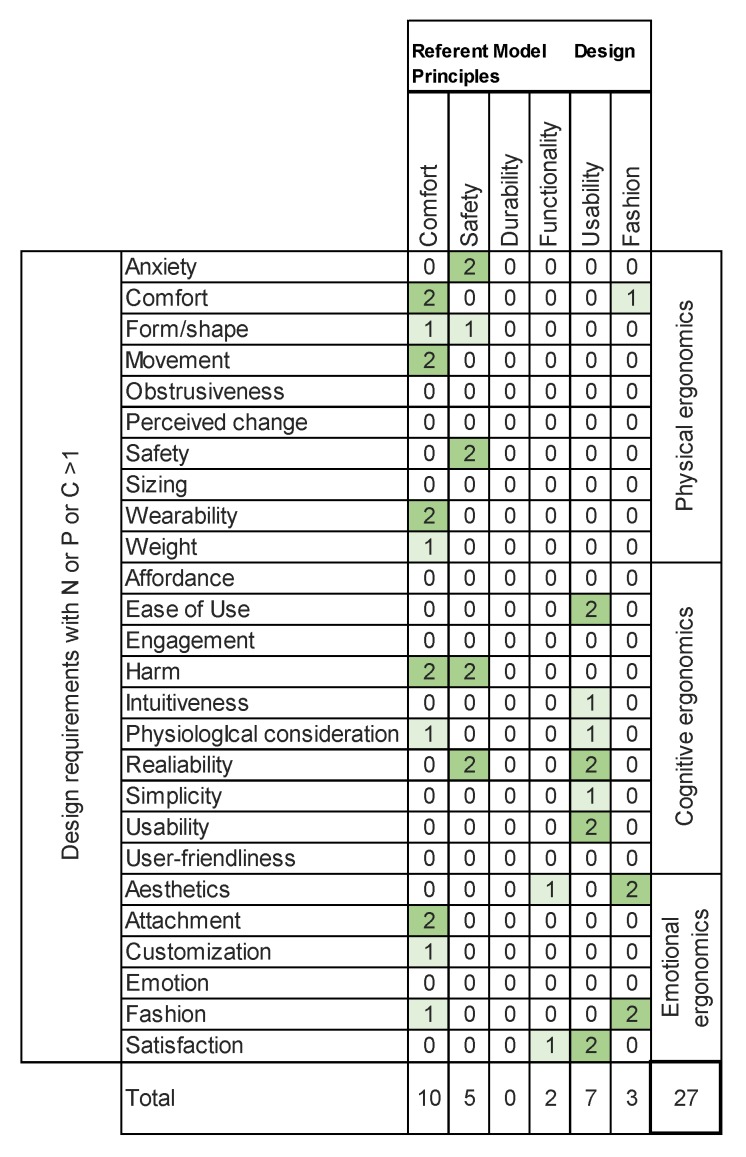
Asymmetric clustering matrix.

**Figure 10 sensors-20-02599-f010:**
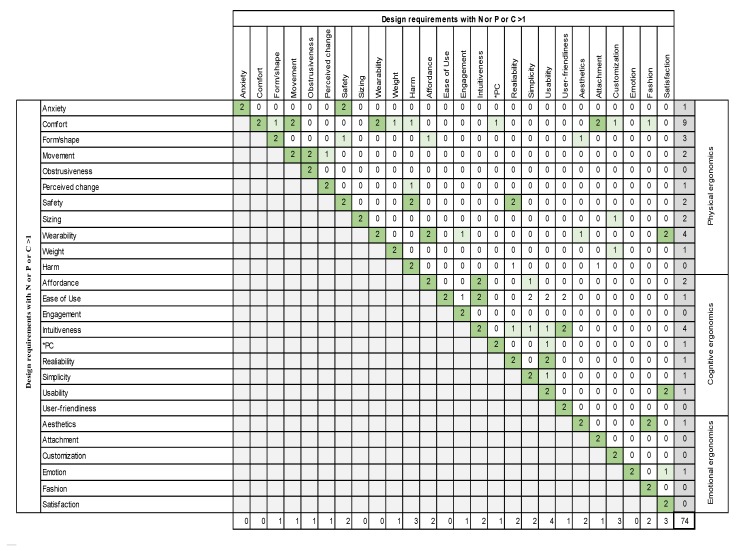
Symmetric clustering matrix.

**Table 1 sensors-20-02599-t001:** Authors’ key messages identified in the literature review.

Author	Title	Key Message
Mayol et al. [9]	“Towards Wearable Active Vision Platforms Positioning the sensor”	Development of a specific wearable through the understanding of the interaction of human body and the wearable.
Knight et al. [13]	“Assessing the Wearability of Wearable Computers”	Wearables, understood as perceptions of wellbeing through comfort assessment.
Dunn, [7]	“Wearability in Wearable Computers”	Wearables understood through the relationship between a worn technology and the ability or desire of the user to wear it.
Pantelopoulos and Bourbakis, [8]	“A Survey on Wearable Sensor-Based Systems for Health Monitoring and Prognosis”	Development of a specific wearable through the understanding of the interaction of human body and the wearable.
Motti and Caine, [6]	“Human Factors Considerations in the Design of Wearable Devices”	Wearables, understood through the understanding of the interaction between the human body and the wearable device.
Contreras-Vidal, [15]	“Human- Centered Design of Wearable Neuroprostheses and Exoskeletons”	Human-centered approach to improve interaction and to make wearables more effective, reliable, safe and engaging.
Tomberg et al. [19]	“Applying Universal Design Principles to Themes for Wearables”	Application of Universal Design Principles to wearables design.
Andreoni et al. [20]	“Defining Requirements and Related Methods for Designing Sensorized Garments”	Non-intrusive monitoring paradigm in order to do not affect user behavior in wearables.
Bhömer, [16]	Designing Embodied Smart Textile Services The role of prototypes for project, community and stakeholders	Exploration of wearables how close-to-the-body products and services can become meaningful to people.
Cho, [5]	“Review and Reappraisal of Smart Clothing”	Wearable Requirements identification through a human view based on Maslow’s hierarchy of needs.
Tomico et al. [17]	“Soft, embodied, situated & connected: enriching interactions with soft wearables”	Designing soft wearables applying notions of personal-meaning–making
Canina and Ferraro, [14]	“Biodesign and human body: a new approach in wearable devices”	Wearables understanding based on the consideration of psycho-physical wellbeing of users.
Ferraro et al. [10]	“Wearability and user experience through user engagement: The case study of a wearable device”	User-centered approach for the understanding of wearability.

**Table 2 sensors-20-02599-t002:** Definition of the terms used for this study.

Legend	
Design Requirements	These are the groups of requirements/principles proclaimed as such by the original study
Definition	This is the definition of the requirement provided by the original study.
Parameter	Other aspects that are not design requirements, but do have an influence on their measurement or validation.

**Table 3 sensors-20-02599-t003:** Quantitative and qualitative dimensions defined by Hignet and Wilson (2004) [23] that will be applied to classify the design requirements as quantitative, qualitative or combined (quantitative+qualitative).

Qualitative Dimensions	Quantitative Dimensions
Words, understanding	Numbers, explanation
Purposive sampling, inductive reasoning	Statistical sampling, deductive reasoning
Social science, soft, subjective	Physical sciences, hard, objective
Practitioner as a human being to gather data, personal	Researcher, descriptive, impersonal
Inquiry from the inside	Inquiry from the outside
Data collection and data intertwined	Data collection before analysis
Creative, acknowledges of extraneous variables as contributing to the phenomenon	Predefined, operationalized concepts stated as hypotheses, empirical measurement and control of variables.
Meanings of behaviours, broad and inclusive focus	Cause and effect relationship
Discovery, gaining knowledge, understanding actions	Theory/explanation testing development

**Table 4 sensors-20-02599-t004:** Classification of design requirements within quantitative, qualitative or combined parameters.

		No of Quantitative Parameters	No of Qualitative Parameters	No of Combined Parameters
**R1**	Comfort	3	1	4
**R2**	Safety		1	1
**R3**	Durability	1		
**R4**	Usability			2
**R5**	Reliability	1		1
**R6**	Aesthetics		3	
**R7**	Engagement		1	1
**R8**	Privacy			1
**R9**	Functionality			1
**R10**	Satisfaction			1
	**Total**	**5**	**6**	**12**

**Table 5 sensors-20-02599-t005:** Suggested assertions from wearables evaluation based on human–computer interaction (HCI) models.

Assertions from HCI Evaluation	Suggested Assertions from Wearables Evaluation
A1: Evaluation is integral to the process of design.	A1: The wearables evaluation starts at the beginning of the design process (in the case of this paper, it starts at the discovery phase).
A2: Evaluation measures more than a product’s features—the preceding discussion has highlighted that evaluation involves the user’s response and the manner in which they use the product.	A2: The wearables evaluation combines quantitative and qualitative methods and the user is involved from the beginning of the design process.
A3: Evaluation implies critique.	A3: The wearables evaluation implies iterative design through different prototypes.
A4: Evaluation involves measurement.	A4: The wearables evaluation involves a combination of measurements and expert analysis.
A5: Evaluation requires a comparison with a reference model.	A5: The wearables evaluation requires a comparison with a reference model.

**Table 6 sensors-20-02599-t006:** Design requirements and suggested evaluation tools.

Design Requirements	Improved Evaluation Proposal
R1 Comfort Shape, Breathability Hygiene Temperature Sizing Obtrusiveness Movement Weight	Washing experiments Multi-layering experiments Intermediate comfort user tests Questionnaires in the first design stage
R2 Safety Harm Anxiety	User test in real scenario
R3 Durability Resistance	User test in real scenario Analysis of the estimated life cycle with the user
R4 Usability Intuitiveness Simplicity	Additional user test in real scenario
R5 Reliability Precision Effectiveness	Real scenario exploration User test
R6 Aesthetics Fashion Form language	Photo-based survey Form language definitions through meetings and focus groups
R7 Engagement Long-term use Engagement	User observation Data analysis
R8 Privacy Privacy Subtlety	Specification definitions in meetings Data analysis User test Co-design
R9 Functionality	Partial calibration experiments Full system calibration experiments User test
R10 Satisfaction	User observation over time User test Questionnaires Co-design

**Table 7 sensors-20-02599-t007:** Questions formulated for the expert survey.

Question	Format
Q1 Experience in Wearables Design *Q 1.1 Have you ever designed wearables?* *Q 1.2 Which kind of wearables have you designed?* *Q1.3 Which are the applications for the designed wearables?*	- Open questions - Closed questions
Q2 Which are the wearable design requirements? *Q 2.1 A list of design requirements is suggested below. Would you eliminate any of them? Choose those which you would eliminate* *Q2.2 Would you add any design requirement?*	- Checklist - Closed questions - Open questions
Q3 Below, design requirements and specific parameters are shown. *Q 3.1 In your view, should we take following parameters into account?* *Q 3.2 From those requirements that you would eliminate, why do you think they should not been considered?* *Q 3.3 Other comments*	- Checklist - Closed questions - Open questions
Q4 How could we evaluate design requirements?	- Multiple choice answer sheet

**Table 8 sensors-20-02599-t008:** Experts’ responses to the consideration of wearable design requirements.

		D1	D2	D3	D4	D5	D6	D7
**R1**	Comfort	•	•	•	•	•	•	•
**R2**	Safety	•	•	•	•	•	•	•
**R3**	Durability		•	•	•	•	•	•
**R4**	Usability	•	•	•	•	•	•	•
**R5**	Reliability	•	•	•	•	•	•	•
**R6**	Aesthetics	•		•	•	•	•	•
**R7**	Engagement		•	•	•	•	•	•
**R8**	Privacy	•	•		•	•	•	•
**R9**	Functionality	•	•			•	•	•
**R10**	Satisfaction	•	•	•	•	•	•	•

**Table 9 sensors-20-02599-t009:** Results obtained from the analysis of the co-evaluation checkpoint application.

Group	Degree of Fulfillment of the Design Requirements in the First Round	Degree of Fulfillment of the Design Requirements in the Second Round	Design Stage Influenced by the Iteration
GR1	2.52	3.43	Brainstorming, Prototype, Service Blueprint, Concept Design
GR2	3.54	4.3	Prototype, Service Blueprint, Concept Design

**Table 10 sensors-20-02599-t010:** Matrix Scoring Criteria.

Score	Score Strength	Score Description
**0**	Weak relation	N or P or C parameters >1
**1**	Medium relation	Items are connected by N or P parameters
**2**	Strong relation	Items are connected by N and P parameters

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
