# Peer review of "Wearable Design Requirements Identification and Evaluation"

_sensors, 2020, doi:10.3390/s20092599_

Round 1

Reviewer 1 Report

It would be useful to have a identified structured list of design requirements for the developing area of wearables, whilst this paper covers the background research I did not find it that clear as to what the recommended design requirements were that were reached after the research, to this end I would like to see the conclusions section expanded to give a clear set of structured requirements and guidance on how and when to apply them during the design process so that other workers in the area can follow the proposed approach.

In general the paper is presented in clear English with no langague issues, there are some minor corrections I would like to see in addition to the expansion of the conclusions section (as above) which are detailed below and indexed by line number in the review pdf.

Line 9; missing word “make it possible”

Line 96; reword “Although there few studies” for clarity

Line 111/112; reword “This fact involves different contexts of use, this study have exclude them for this theoretical framework.” for clarity

Fig. 2; Text in the figure is not very clear / legible in review copy, please check/redraw

Line 116/118; Fig. 2 only appears to have 7 studies not the stated 8, please check and correct

Line 147; “Number of time Error s the requirement” error in pdf generation, please correct

Line 170/171/172; “Regarding the number of times "comfort" appears in another definition, it can be seen that there are eight definitions including comfort” This is statement is unclear from the figure, please rewrite to explain better

Line 173/174; “Figure 3, Figure 4 and Figure 5 show the quantification of the N, P and C parameters for the list 173 of design requirements organized according to the three ergonomic domains” please make it clearer what the information in these figures tells the reader

Line 206; Please include the reference number for Cho here

Line 255/257; Author previously referred to as Cho is now Gilsoo Cho? Please make references consistent throughout

Line 293-295; “The results show that through the reviewed tools, we cannot assure the design requirements evaluation. Following section will analyse the measurability of design requirements and then, other fields will be studied” It is not clear what the results are in the table, this section needs improvement so the reader knows what is being evaluated and what it means

Fig 10; rewrite caption to include refences(s) to source material. “Dimensions of Qualitative and Quantitative Methodologies. Source. Evaluation of Human 306 Work Book Source Evaluation of Human Work.” Also, should this be a table as opposed to a figure?

Line 327/332; Please define HCI acronym after full text version, before three-letter version is used.

Line 335-340; This is a direct repeat of lines 329-334, please delete.

Table 4; this table is hard to follow, please consider separating the ‘R numbers’ from the properties. Eg from:

 “R2 Safety

Harm

Anxiety”

To

“R2:

Safety

Harm

Anxiety”

Line 380; Please give more reference details for “International Design Conference (Dubrovnik)”

Line 383; it is not clear what “Level 2” and “Level 3” are referring to in this context, please clarify.

Line 411; Repeated section heading number 6.1

Table 6; This table does not seem to be referred to in the text.

Line 449; Please remove stray single bullet point

Line 455; Reference to figure 1 seems incorrect.

Line 474-477; It is unclear what this section is relating to, please re-write.

Line 497-498; The proposed framework needs to be summarised in the conclusions section.

Line 501-503; Please summarise which requirements require which methods.

End of comments

with kind regards

Author Response

First of all, we would like to thank the different modifications suggested that have contribute to improve the paper. We have modified the different specific changes proposed line by line and we have also modified some sections. The introduction and conclusions of the paper have also been reformulated based on the different comments made by the reviewers. In this last version, the significance, novelty challenges and contributions of the papers are better structured and empathized.

As far as the English is concerned, it has been internally checked with other researchers. Due to the current situation we have not been able to send it to an additional professional checking within 10 days but we are able to send it later if it would be required to.

Reviewer 2 Report

1. The paper by L. Francés, et al. describes the identification and assessment of the necessities when designing wearable devices/systems. The current manuscript is quite vague. The writing and organization must be improved. Many revision points are required. After major careful revision, if the authors provide detailed discussion and clarify the points, it might be interesting to reconsider.
2. The authors wrote “Why do these high-tech devices have such life cycle problems?” at the end of the first paragraph. However, note that a thesis statement is one sentence that expresses the main idea of a research paper. It is unclear and does not get into the main point.
3. Line 14-15. Please add the journal/year/or DOI for “Design and Development of a Low-Cost Wearable Glove to Track Forces Exerted by Workers in Car Assembly Lines and published in Sensors”
4. Line 18. Please clarify which processes that authors could help provide a roadmap for wearable designers and researchers
5. “human factors” which are mentioned in the abstract should be elaborated (It could be clearly in the introduction). Currently, the phrase “human factors” is dangling and vague. The authors should exemplify or define clearly which factors.
6. Not sure what is the criteria to mention “they were the first authors that included the term “wearability” (Line 39-41). Considering that many people used “wearability” (such as with wearable devices) earlier than 2014? For example, 10.1109/TSMCC.2009.2032660; 10.1109/ISWC.2008.4911605; 10.1109/ICSMC.2000.886255 ; and many more.
7. The authors should state more clearly the novelty of this research effort. What is unique, novel, or better about your work as compared to the existing art of literature? The table, which includes the summary and comparison of this current work versus other reports, should be added.
8. In the introduction, please state explicitly:
8.1. Significance
8.2. Novelty
8.3. Challenge/research questions
8.4. Scope
8.5. Key achievement/new findings
8.6. What will you offer to solve the existing problem? (The answer to this research question can be your thesis statement.)
9. Figure 2 should be reorganized. It is should be analyzed the categories and similarities clearly. Some “cognitive”, “physical”, “emotional”, etc. could be grouped. Currently, it is very crowded and too fragmental; contains no connections, and not comprehensive. It should show a good and clear graphical concept for visualizing connections between several ideas or fragments of information. A web of clear relationships should be designed.
10. For ‘Conclusions’, the challenges of this research that the authors have addressed should be emphasized. How do your effort and new findings address exiting art?
11. For ‘Conclusions’, please specify the interesting points over other reports or clearly conclude the new and significant findings.
12. This article would be improved by careful editing. Some stylistic and grammatical errors (languages) should be rechecked. It would help the readers to follow the authors’ argument.

Author Response

(The authors gave the same response as above.)

Reviewer 3 Report

The present paper is a complex analysis of the requirements for wearables design that, for the first time, provides a strong analysis of the topic taking into account more "human" factors than normally done, with the actual literature just mostly facing more "technical" and "technological" concerns.

The analysis performed is quite complex and elaborated, therefore it is quite difficult to summarize it and to draw a highly readable paper.

In my opinion, the approach used is quite correct from a methodological point of view, therefore my concerns are mainly related to the presentation quality, with some tips as summarized below.

Section 2

Figure 2 is quite hard to follow. I see you have categorized the bubbles into design or group requirements, which is methodologically correct but honestly difficult to read and quite confusing to my eyes. Please, consider also different ways of presentation. In addition, it seems that just seven studies are reported there, whereas the text mentions eight of them. Please, check.

Section 6

The paragraph 6.1. Survey Results should be renumbered, as the 6.1. was already assigned to the paragraph “Survey”.

Section 7

I would restructure this section in order to improve its readability and its logical flow.

Overall, language and grammar should be revised throughout the manuscript with the support of a native speaker. Also, typos should be corrected.

Author Response

(The authors gave the same response as above.)

Round 2

Reviewer 1 Report

Thank you for making many of my suggested changes, and for adding the additional text to the conclusions. I still have a few other minor comments to be addressed which are as below:

Line 41,46,54,56,68,70 these references include the year, whilst others do not, please make the referencing style consistant throughout

line 65, reference number missing

Throughout the document references to Cho vary between Gilsoo Cho and Cho. Please search for and correct this so that they all say Cho.

Line 354 ‘que quantitative’ ? Please check

Line 378 change ‘Human Computer’ to ‘Human-Computer’ for consistency with lines 371, 373

Line 384, A1. In table Consider changing ‘thesis’ to ‘paper’

The additional text in the conclusions section would benefit from being re-read and checked for English clarity by another person if possible to improve the readability, though I appreciate that this may be difficult at this time.

kind regards

Author Response

First of all, we would like to thank the different modifications suggested that have contribute to improve the paper. We have modified the different specific changes proposed line by line and we have also modified some sections. Due to this reason, some of the lines may have considerably changed.

Firstly, different specific modifications have been done following the recommendations of the reviewers (these modifications are explained line by line below). Secondly, based on the comment of some reviewers some structural changes have been done; introduction section has been divided in two different sections and methodological details have been moved to a new section named 9. Supporting Material. Finally, some English corrections have also been done.

Reviewer 2 Report

1. The authors have revised the manuscript considering suggestions and comments. The paper has a good point that the authors analyze the requirements for wearables design. However, the quality may not reach to publish in this level of original research of this Sensors Journal yet. The organization must be critically improved. After the re-submission, the reviewer would be happy to review the revised article again.
2. It is a necessity that the authors take more careful time to reorganize the whole outline and writing the paper. It is not due to English, but the paper is not clearly structured yet. The authors may wish to reconsider to move some data to the Supporting Information, and enhance the quality of the main and significant data (and key Figures). In addition, the writing should be concise. Some methodological details can be moved to Supporting Information, then highlighting the focus and key.
3. The abstract should be an independent document that can be communicated on its own. This makes the inclusion of references/citations problematic. Citing “[1]” is not suitable.
4. Line 14 (abstract). It should clarify the writing that “an empirical case study” was already reported.
5. Please recheck critically. It seems that there are references that already discussed “the term “wearability” understood as the interaction between the human body and the wearable object” prior to this [Ref. 7].
6. Fig. 2 is still unorganized, crowded and fragmental. It should not be just throwing information to the map without critical digestion, analysis, and presentation. Currently, it does not show visualizing connections between several ideas or fragments of information.
7. Many Data and Figures are not clearly visible. For example, some texts in Figure 8 can be enlarged.
8. The literature review and similar papers/analyses can be structured in a form of Table to highlight the point of the key messages (get to the point).

Author Response

(The authors gave the same response as above.)

Reviewer 3 Report

The paper was significantly improved with respect to the previous round.

I just remind to perform a language and grammar revision by a native English speaker, since there are still several mistakes, as well as typos, throughout the text.

Author Response

(The authors gave the same response as above.)

Round 3

Reviewer 2 Report

1. The authors have revised the manuscript considering suggestions and comments. It is better toward publication. Please, additionally, address these in the final manuscript:
2. Fig. 2. Please improve visibility and contrast. Please consult some guides: "A brief guide to designing effective figures for the scientific paper." Advanced Materials 23.38 (2011): 4343-4346.; "Ten simple rules for better figures." PLoS Comput Biol 10.9 (2014): e1003833.
3. Please consider ordering the papers in Table 1 systematically. (Probably by year)
4. Table 1. The additional column showing the title of each paper should be helpful for the reader.
5. Table 13-14. Please consider that the table is long and extended to more than one page. The head (QN QL C) will be missing and cause difficulty for the reader to follow. Please improve. Using distinct colors is also helpful. The authors may also wish to encode the cells in the table by the attentive attribute of color.
6. How did the authors order the info in Table 11 (from the top row to bottom row)? Please consider doing systematically.

Author Response

We would like to thank the revisions and modifications that have contribute to improve this paper. We have modified the different specific changes proposed line by line and the answers are shown in the attached document as well as in the final manuscript where the option “Track Changes” has been used.